# OpenReview forum: "GeoCMON: Operator Learning on Deformable Domains via Disentangled Geometric Conditioning"
_ICLR.cc/2026/Conference — ICLR 2026 Conference Withdrawn Submission_

### Official Review · Reviewer_7jRT · 2025-10-19

**Soundness:** 2
**Presentation:** 3
**Contribution:** 3
**Rating:** 6
**Confidence:** 3

**Summary:**

Existing neural operator methods perform well on fixed geometric domains, but their generalization ability and training stability degrade significantly when partial differential equations are defined on deformable or non-rigid geometries. To address this limitation, this paper aims to design a network architecture capable of stably and accurately learning PDE solution operators on deformable domains, thereby achieving robust operator learning under complex geometries and heterogeneous boundary conditions. The authors attribute the failure of existing approaches on deformable domains to the entanglement between geometric features and boundary conditions, and therefore propose a multi-branch disentanglement architecture that independently encodes geometric deformations and boundary conditions to achieve feature separation and stable modeling. In addition, the paper introduces a physics-aware weighted loss function that directs the model to focus more on physically significant high-magnitude regions during training, improving predictive accuracy and enhancing optimization stability.

**Strengths:**

1. Clear motivation. This paper identifies a well-defined and underexplored challenge in neural operator learning.
2. Principled architectural design. This paper proposes GeoCMON that explicitly disentangles geometry and boundary representations, addressing the core cause of representational entanglement.
3. Physics-aware learning objective. The magnitude-aware weighted loss function is physically meaningful, emphasizing high-importance regions and improving convergence behavior.
4. Comprehensive empirical evaluation. This paper provides systematic experiments covering accuracy, feature orthogonality, training dynamics, and robustness to geometric perturbations, demonstrating consistent and significant improvements over the baseline.

**Weaknesses:**

While the paper frequently refers to “deformable domains parameterized by complex, evolving topologies” (Sec. 1, Line 35) and “manifold-evolving domains” (Sec. 3, Lines 125–127), the experimental evaluation is restricted to steady-state 2D Laplace problems with static, non-rigid geometries generated via mesh perturbations. Consequently, the results primarily demonstrate the model’s robustness to geometric and boundary variations, rather than its ability to handle genuinely evolving or time-dependent domains.

In addition, the paper lacks a deeper theoretical foundation explaining why disentangling geometry and boundary representations should improve generalization. The current argument is mainly empirical.

**Questions:**

1. Has GeoCMON been evaluated on nonlinear / time-dependent / 3D PDEs? If not, could the authors provide additional experiments or discussion regarding the model’s applicability and potential challenges in such settings?
2. In Sec. 4.1 (Datasets), the authors note that each 2D mesh typically contains around 40 spatial nodes. This resolution appears relatively coarse. Have the authors conducted experiments on higher-resolution datasets to examine how model accuracy and computational cost scale with finer spatial discretizations?

---

### Official Review · Reviewer_RpKY · 2025-10-21

**Soundness:** 2
**Presentation:** 1
**Contribution:** 1
**Rating:** 2
**Confidence:** 4

**Summary:**

This work considers the problem of learning neural operators on complex geometries. The authors propose to use an autoencoder architecture to map the complex geometry to a latent space, where a neural operator (in this case a DeepONet) is trained to learn the mapping between input and output functions. The method is tested on a 2D Laplace problem defined on complex geometries, demonstrating its ability to generalize across different shapes.

**Strengths:**

- Being able to learn neural operators on complex geometries is an important problem in scientific machine learning and many methods struggle to impose constraints like boundary conditions on complex domains.
- The idea of using an autoencoder to map complex geometries to a latent space is interesting and could potentially allow for better generalization across different shapes.

**Weaknesses:**

- The main contribution of the paper is very unclear, especially the methodology section (Section 3). As far as I understand, it seems that the proposed method is an autoencoder architecture, mapping the geometry to a latent space, with a neural operator acting on the latent space (in this case chosen to be a DeepONet). I do not see the difference between this approach and the geometry-aware FNO proposed by Li, Huang, Liu & Anandkumar, beyond the change of the FNO by a DeepONet. The authors should clarify what is the novelty of their approach compared to existing methods.
- The introduction section and abstract are very vague about the actual contributions of the paper. Terms such as representational entanglement are not really defined or explained.
- There is a lot of formalization in Section 3.3 for expressing a trivial fact that one uses an $L^2$ norm for training the model, where the weights correspond to quadrature weights for approximating the integral. This is standard practice in operator learning, and does not require a lengthy formalization.
- The applications considered (2D Laplace problem) are extremely simple and do not showcase the advantage of using a neural operator. There is also no comparison with other neural operator methods acting on complex geometries such as Geo-FNO or GraphNO.

**Questions:**

- How does the method compare with other neural operator methods designed for complex geometries, such as Geo-FNO or GraphNO?
- What is the main novelty of the proposed method compared to existing geometry-aware neural operators?
- How does one choose the ``physics-aware loss function'', i.e. the weight parameters in the loss?
- Why do the authors need to perform dimensionality reduction on the dataset given the simplicity of the considered problem?

---

### Official Review · Reviewer_qfWr · 2025-10-29

**Soundness:** 2
**Presentation:** 2
**Contribution:** 2
**Rating:** 2
**Confidence:** 3

**Summary:**

The authors propose GeoCMON, a neural operator architecture designed to learn PDE solutions on complex geometries, with potentially deformable domains and evolving topologies. The core idea of the paper is to address the representation entanglement problem. Existing architectures address it via simple fusion strategies. Instead, the model take inspiration from the DeepOnet work and use a multi-branch architecture, combining a branch that encodes a PCA-based representation of the domain geometry, a second encoding the discretized boundary conditions and a trunk network encoding the spatial coordinates. Feature fusion is performed via a Hadamard product and tensor contraction. They also introduce residual connections to stabilize training and a magnitude-aware loss to help focus on high-magnitude regions of the solution.

Experiments are performed on the 2D Laplace problems.

**Strengths:**

The paper tackles a clear and significant challenge for neural operators applied to scientific data. Many neural operators often fail to generalize to varying domain geometries.

The core architectural choice seems intuitive.

Different analysis are performed to provide a comprehensive evaluation (analyses of feature orthogonality, training dynamics, gradient noise, stability to geometry perturbations).

**Weaknesses:**

If my comprehension of the results is correct, the paper's central claims are, in several places, directly and unambiguously contradicted by the results presented in the different figures/tables.

In appendix B, the text claims the weighted loss improves the accuracy in critical regions and that it achieves a lower error in low magnitude regions that the baselines. Table 3 seems to show that the baseline has a lower L2 and RelL2 error in all three regions.

In appendix D, again it is stated that the model better adapt than the baselines at stages 2 and 3, but the figure 13 seems to show the opposite.

Concerning the novelty of the paper, the tri-branch directly take inspiration from the DIMON baseline. The conditional residual connection is just a classical residual block. Finally, the fusion technique is a minor modification of the classical dot product fusion technique used in DeepONets.

The experiments are only performed on the 2D Laplace equatin, a simple linear elliptic PDE.

**Questions:**

Can the authors provide any results on more challenging problems (non-linear, time-dependent, or 3D)?

How does the method scale with respect to mesh resolution and geometric complexity that requires a much larger PCA basis?

**Details Of Ethics Concerns:**

No ethical concerns.

---

### Official Review · Reviewer_6FLk · 2025-11-01

**Soundness:** 3
**Presentation:** 3
**Contribution:** 3
**Rating:** 6
**Confidence:** 2

**Summary:**

This paper introduces GeoCMON (Geometric-Conditioned Multi-Branch Operator Network), a neural operator framework for learning PDEs on deformable and parametrically varying domains. The work targets a key limitation of existing operator-learning approaches—the entanglement between geometry, boundary conditions, and solution behavior, which often leads to instability and poor generalization on non-rigid domains. Empirical evaluations on 2D Laplacian problems across various deformations and boundary configurations demonstrate that GeoCMON achieves superior accuracy and smoother optimization dynamics compared to the DIMON and other baselines. The paper also provides gradient noise and feature orthogonality analyses, offering insight into optimization stability and representation disentanglement.

**Strengths:**

The paper presents a well-motivated and technically solid framework that explicitly disentangles geometric and physical representations in operator learning. This factorization approach effectively addresses the challenge of generalization across deformable domains.
GeoCMON achieves consistent improvements over the DIMON baseline, exhibiting lower test errors and enhanced stability in training. The authors go beyond traditional performance comparisons by examining optimization behavior—such as gradient variance and synchronization—providing a deeper empirical understanding of why GeoCMON performs better.
The introduction of Conditional Residual Blocks improves gradient propagation, while the magnitude-aware weighted MSE loss helps balance training across varying scales of solution magnitude, leading to smoother convergence.
The model is also conceptually interpretable: separating the geometry and boundary-condition encoders is a principled way to model PDEs defined over non-rigid or evolving topologies, and the proposed fusion mechanism elegantly integrates their information.
Overall, the paper contributes both architectural insight and empirical rigor, offering a clear step forward for operator learning on complex geometric domains.

**Weaknesses:**

While the paper’s design is sound and its results are convincing, its novelty is somewhat limited. The proposed architecture closely resembles DIMON, which already introduced a multi-branch design (two encoders plus a trunk). GeoCMON’s main innovations—conditional residual connections and the magnitude-weighted loss—represent incremental but meaningful refinements rather than a fundamental shift.
Second, the experimental scope is narrow: the paper evaluates only on 2D Laplace problems. Given that the motivation emphasizes deformable and heterogeneous domains, it would be valuable to demonstrate results on other PDE types (e.g., nonlinear elliptic or parabolic equations) to confirm broader applicability.
Additionally, although the authors provide heuristic explanations for the advantages of the disentangled architecture and training dynamics, the lack of formal theoretical justification leaves open questions about why the model generalizes better beyond empirical evidence.
Finally, the discussion on dimensionality reduction (e.g., PCA-based representation) could be expanded. It is unclear whether PCA is optimal for geometry encoding or whether other techniques (e.g., autoencoders, manifold embeddings) could yield stronger representations.

**Questions:**

- Could GeoCMON be extended to handle time-dependent PDEs on time-varying or non-rigid domains, such as reaction–diffusion or Navier–Stokes equations?
- Have the authors considered alternative dimensionality reduction methods (e.g., autoencoders or kernel embeddings) for constructing geometry or boundary representations beyond PCA?
- How sensitive is the model to the choice of weighting scheme in the magnitude-aware loss? Could it be adapted dynamically during training for additional stability?
- Is the Einstein-summation-based fusion mechanism generalizable to higher-dimensional domains or unstructured meshes?

---

### Note · Authors · 2025-11-12

I have read and agree with the venue's withdrawal policy on behalf of myself and my co-authors.